# Identical anthropometric characteristics of impaired fasting glucose combined with impaired glucose tolerance and newly diagnosed type 2 diabetes: anthropometric indicators to predict hyperglycaemia in a community-based prospective cohort study in southwest China

Fang Zhang,[1] Qin Wan,[2] Hongyi Cao,[3] Lizhi Tang,[1] Daigang Li,[4] Qingguo Lü,[1] Zhe Yan,[1] Jing Li,[1] Qiu Yang,[3] Yuwei Zhang,[1] Nanwei Tong[1]

For numbered affiliations see end of article.

**Correspondence to**
Dr Nanwei Tong;
tongnw@scu.edu.cn

## ABSTRACT

**Objectives** To assess the anthropometric characteristics of normoglycaemic individuals who subsequently developed hyperglycaemia, and to evaluate the validity of these measures to predict prediabetes and diabetes.

**Design** A community-based prospective cohort study.

**Participants** In total, 1885 residents with euglycaemia at baseline from six communities were enrolled.

**Setting** Sichuan, southwest China.

**Primary outcome measures** The incidences of prediabetes and diabetes were the primary outcomes.

**Methods** The waist-to-height ratio (WHtR), body mass index (BMI), waist circumference (WC) and waist-to-hip ratio (WHR) of all participants were measured at baseline and during follow-up. A 75 g glucose oral glucose tolerance test was conducted at each survey.

**Results** During a median of 3.00 (IQR: 2.92–4.17) years follow-up, the cumulative incidence of isolated impaired fasting glucose (IFG), isolated impaired glucose tolerance (IGT), IFG combined with IGT (IFG+IGT), and newly diagnosed diabetes mellitus (NDDM) were 8.44%, 18.14%, 8.06% and 13.79%, respectively. WHtR, BMI, WC and WHR were significantly different among subjects who subsequently progressed to isolated IFG or IGT, IFG+IGT or NDDM (p<0.05). The anthropometric characteristics of IFG+IGT subjects were similar to those of the NDDM population (p>0.005). All the baseline anthropometric measurements were useful for the prediction of future prediabetes and NDDM (p<0.05). The optimal thresholds for the four measurements were calculated for the prediction of hyperglycaemia, with a WHtR value of 0.52 performing best to identify isolated IFG or IGT, IFG+IGT and NDDM.

**Conclusions** Anthropometric measures, especially WHtR, could be used to predict hyperglycaemia 3 years in advance. Distinct from isolated IFG and IGT, the

## Strengths and limitations of this study

► This study described and compared the anthropometric characteristics of participants who subsequently progressed to isolated impaired fasting glucose (IFG), isolated impaired glucose tolerance (IGT), IFG combined with IGT, newly diagnosed diabetes mellitus (NDDM) or who remained normoglycaemic.

► Variations in waist-to-height ratio, body mass index, waist circumference and waist-to-hip ratio were used to predict the transition from euglycaemia to prediabetes, and overt NDDM in the following 3 years.

► The optimal threshold values for the prediction of hyperglycaemia were determined from the anthropometric measurements collected.

► The inherent limitations of the present work were a relatively short follow-up period (median 3 years), a low completion ratio of 41.9% and a limited sample size, meaning that anthropometric threshold values could not be determined by gender for each category of hyperglycaemia.

individuals who developed combined IFG+IGT had identical anthropometric profiles to those who progressed to NDDM.

## INTRODUCTION

The rapidly growing incidence of diabetes means that it is now reaching epidemic proportions in China. The overall prevalences of diabetes and prediabetes were estimated to be 11.6% and 50.1%, respectively, in Chinese adults in 2010.[1] In 2007–2008,

another cross-sectional study conducted across China found that the prevalences of isolated impaired fasting glucose (IFG), isolated impaired glucose tolerance (IGT) and IFG combined with IGT (IFG+IGT), were 3.2%, 11.0% and 1.9% in men, and 2.2%, 10.9%, and 1.7% in women, respectively.[2] Isolated IFG, isolated IGT and IFG+IGT were selected as three different categories of prediabetes, reflecting the progression from euglycaemia to type 2 diabetes (T2D). Approximately 75%–80% of patients with diabetes develop cardiovascular disease (CVD) ultimately, and patients with prediabetes have also been shown to be at greater risk of heart attack and stroke.[3–5] It has been estimated that between 2005 and 2015, diabetes and consequent CVD have cost China US$557.7 billion.[6]

Measures to limit prediabetes are critical for the prevention of diabetes. Early recognition of prediabetes and prompt intervention could reduce the impact on society as a whole. Both overall and central adiposity are closely linked to hyperglycaemia. Body mass index (BMI) correlates with overall adiposity, while waist circumference (WC), waist-to-height ratio (WHtR) and waist-to-hip ratio (WHR) are indicators of central obesity. These four anthropometric indices are used globally to assess the risk of current or future diabetes.[7–9]

Anthropometry is an affordable and practical screening tool for the presence of hyperglycaemia, in both wealthy and impoverished areas of China. In this community-based prospective cohort study, we aimed to determine whether these anthropometric indices could predict future prediabetes and diabetes, and to establish optimal threshold values for the population. The baseline anthropometric characteristics of normoglycaemic subjects, who subsequently developed isolated IFG, isolated IGT, IFG+IGT and newly diagnosed diabetes mellitus (NDDM) during follow-up, were compared and the similarities and differences between pairs of hyperglycaemic categories were analysed.

## STUDY DESIGN AND METHODS
### Study population
The present study included two populations, in Luzhou City and in the Wenjiang area of Chengdu City. The Luzhou population are participants in the Risk Evaluation of cAncers in Chinese diabeTic Individuals: a lONgitudinal (REACTION) study which is a multicentre prospective observational study of 25 communities in mainland China.[10 11] A total of 10 007 residents, aged 40–89 years, were randomly recruited to participate in this study from five communities in Luzhou in 2011. Subjects with a history of diabetes, incident diabetes or prediabetes verified by an oral glucose tolerance test (OGTT), those missing values or any parameter, or having any of the other conditions (listed below), were excluded. After this, 3800 individuals with normoglycaemia remained to form the baseline population. Of these, 1354 participants returned to complete the study

in 2014. In addition, in 2016, 228 members of the baseline normoglycaemic population who had not been studied in 2014, were followed up. Therefore, data from a total of 1582 subjects from Luzhou baseline screen were available for analysis.

In the Wenjiang survey, a cohort of 1104 participants aged 40–75 years were randomly recruited from Yinchao community in 2011. Using the same inclusion criteria, 698 normoglycaemic individuals comprised the baseline population. Of these, 303 subjects were followed up in 2015 and completed the study. Thus, from Luzhou and Wenjiang, a total of 1885 participants were included in the analysis.

All of the subjects were of Han Chinese ethnicity. A flow diagram of the study design is displayed as supplementary figure 1 (online). Individuals with the following conditions were excluded from the study: infection, pregnancy, malignant tumour, acute cardiovascular accident, serious trauma, liver or renal dysfunction, or a long history of glucocorticoid use. The research was conducted in accordance with the principles of the Declaration of Helsinki II. Each participant provided written informed consent.

### Patient and public involvement
All patients were randomly recruited to participate in this study and were interviewed face-to-face by trained investigators for detailed explanation of informed consent at the beginning. Three months later, each participant received a health report with advised suggestions.

### Diagnosis of diabetes and prediabetes
The diagnosis of hyperglycaemic disorder was made in accordance with the American Diabetes Association recommendations, using OGTT, in 2011.[12] Normal glycaemic tolerance (NGT) was defined by a fasting plasma glucose (FPG) <5.6 mmol/L and a 2-hour plasma glucose (2hPG) <7.8 mmol/L. Isolated IFG was defined by 5.6 mmol/L ≤FPG <7.0 mmol/L and a 2hPG <7.8 mmol/L, while isolated IGT was defined by an FPG <5.6 mmol/L and 7.8 mmol/L ≤2hPG <11.1 mmol/L. IFG+IGT was defined by 5.6 mmol/L ≤FPG <7.0 mmol/L and 7.8 mmol/L ≤2hPG <11.1 mmol/L. Diabetes was defined by an FPG ≥7.0 mmol/L and/or a 2hPG ≥11.1 mmol/L.

### Anthropometric measurements
Anthropometric measurements, including body mass, height, WC and hip circumference were made by trained investigators. Measurements were conducted while all participants were wearing light clothing, without footwear after a 10–12 hours overnight fast in the morning. Measurements were made using calibrated weighing scales, standard steel strip stadiometers and tape measures. The results were recorded to the nearest 0.1 kg or 0.1 cm. WC was measured at the midpoint between the costal border and the iliac crest at the end of exhalation. Hip circumference was measured around the widest portion of the buttocks. BMI was calculated as body mass (kg) divided by height squared (m$^2$), WHtR was calculated as

WC (cm) divided by height (cm), and WHR as WC (cm) divided by hip circumference (cm).

## Lifestyle variables and biological evaluation

Trained investigators collected lifestyle information, consisting of demographic characteristics, current smoking status, physical activity situation, medications, and personal and family disease histories, using a standard questionnaire and face-to-face interviews. The questionnaire categorised the participants into two groups: subjects undertaking vigorous physical activity ≥1 day per week and subjects undertaking vigorous physical activity on <1 day per week. Blood pressure was measured three times in each participant using an electronic sphygmomanometer (OMRON HEM-7220; Liaoning, China), with 5 min intervals between measurements, after at least 10 min rest, and the mean value was recorded.

All participants underwent an OGTT. After a 10–12 hours overnight fast, venous blood was drawn both before and 2 hours after they drank 300 mL water containing 75 g anhydrous glucose within 5 min. FPG and 2hPG concentrations were measured within 24 hours using the hexokinase method (Hitachi 7600 automatic biochemical analyser; Hitachi, Tokyo, Japan). Fasting blood samples were collected for lipid profile measurements, including total cholesterol (TC), triglyceride (TG), high-density lipoprotein cholesterol (HDL-c) and low-density lipoprotein cholesterol (LDL-c). Serum TC, TG and HDL-c concentrations were measured using oxidase colorimetric methods, and LDL-c concentration was measured by homogeneous assay, on a Hitachi 7600 automatic biochemical analyser (Hitachi, Tokyo, Japan) within 24 hours. Haemoglobin A1c (HbA1c) was measured using the high performance liquid chromatography (VARIANT II TURBO Haemoglobin Testing System, Bio-Rad Laboratories, California, USA). The samples were stored at −20°C until analysis which was undertaken within 3 weeks.

## Statistical analysis

Data were analysed using SPSS software V.16.0 (SPSS, Chicago, Illinois, USA) and MedCalc software V.15.2.2 (MedCalc software, Ostend, Belgium). All data are expressed as mean±SD, median (IQR) or frequency (%), as appropriate. One-way analysis of variance was used for parametric data, whereas the rank-sum test was applied for non-parametric data. The $\chi^2$ test was used for the comparison of ratio. All tests were two-sided. In analyses of more than three groups, overall $p<0.05$ was considered significant. The Bonferroni correction and $\chi^2$ segmentation were used for multiple comparison adjustments. For the comparison of two specific subgroups, $p<0.005$ was considered significant. For BMI, WHtR, WC and WHR, receiver operating characteristic (ROC) curve analyses were used to compare their ability to predict incident prediabetes and diabetes. The non-parametric approach described by DeLong et al was used to compare the areas under ROC curves.[13] The predictive threshold values

for hyperglycaemia were calculated. Cox proportional hazards regression was used to evaluate associations between anthropometric indices and hyperglycaemic categories; the time axis consisted of the period of follow-up until prediabetes or diabetes developed, or the end of the study. Hazard ratio (HR) and 95% CI were calculated.

## RESULTS

### Characteristics of subjects at baseline

A total of 1885 normoglycaemic subjects (649 men and 1236 women), with a median age of 56 (IQR: 48–61) years old, were recruited in 2011. After a median follow-up of 3.00 (2.92–4.17) years, 159 individuals had developed isolated IFG, 342 had developed isolated IGT, 152 had developed IFG+IGT, 260 had developed NDDM and the remaining 972 participants remained normoglycaemic. The incidences of prediabetes and NDDM were calculated to be 104.9 per 1000 person-years and 41.8 per 1000 person-years, respectively. The characteristics of all the subjects at baseline in Luzhou and Wenjiang are shown in supplementary table 1 (online). The participants in Luzhou were older than the participants in Wenjiang, and had higher glucose levels at baseline and greater incidences of prediabetes and diabetes during follow-up. The baseline measurements of the participants who subsequently developed isolated IFG, isolated IGT, IFG+IGT or NDDM in the future are shown in table 1. The subjects who developed NDDM were the oldest group at baseline among the five groups (p=0.000). The individuals who transited to isolated IGT, IFG+IGT or NDDM had higher baseline HbA1c levels than the subjects who remained normoglycaemic (p<0.005).

### Baseline and follow-up anthropometric values in subjects who subsequently developed hyperglycaemic disorders

During the follow-up examination, it was found that WHtR in the NGT group was lower than in the isolated IGT, IFG+IGT or NDDM groups (p<0.005) (table 2), and it was lower in the isolated IFG and isolated IGT groups than in the IFG+IGT and NDDM groups (p<0.005). The p values were 0.009 and 0.006 for BMI in isolated IFG versus IFG+IGT, and isolated IGT versus IFG+IGT, respectively, and 0.005 for WHR in the isolated IFG or IGT groups versus the IFG+IGT group. There were the trends towards the differences in both BMI and WHR between the isolated IFG or IGT groups, and the IFG+IGT group. To summarise, BMI, WC and WHR in the five hyperglycaemic groups tended to follow the following pattern: NGT < isolated IFG and isolated IGT < IFG+IGT and NDDM. Unlike when the isolated IFG or isolated IGT groups were compared, the anthropometric characteristics of the IFG+IGT group were similar to those of the NDDM at follow-up (p>0.005).

To assess whether the anthropometric values were already different before hyperglycaemia developed, we evaluated the differences between groups at baseline,

**Table 1** General measurements of subjects at baseline who progressed to hyperglycaemia at follow-up

| | NGT (n=972) | Isolated IFG (n=159) | Isolated IGT (n=342) | IFG+IGT (n=152) | NDDM (n=260) | Overall p values |
|---|---|---|---|---|---|---|
| Follow-up time (year) | 3.00 (2.92–4.17)* | 3.00 (2.92–4.17)* | 2.92 (2.92–3.17)† ‡ | 3.00 (2.92–3.17) | 3.00 (2.92–3.17) | 0.000 |
| Age (year) | 53 (46–59)‡ *§¶ | 55 (48–62)†¶ | 59 (49–65)† | 56 (49–62)†¶ | 60 (54–65)†‡§ | 0.000 |
| Female | 675 (69.44%) | 96 (60.38%) | 220 (64.33%) | 97 (63.82%) | 166 (63.85%) | 0.075 |
| Height (cm) | 158.00 (153.10–164.00) | 159.45 (154.00–165.52) | 157.00 (152.00–162.70) | 157.10 (154.00–164.00) | 156.00 (152.00–163.20) | 0.492 |
| Weight (kg) | 58.00 (52.00–65.00) | 60.50 (53.99–66.85) | 60.00 (53.00–66.20) | 62.10 (56.70–69.50) | 62.00 (55.00–69.75) | 0.498 |
| Hip circumference (cm) | 93.00 (88.20–97.20) | 94.00 (90.00–99.00) | 95.00 (90.20–100.00) | 96.00 (92.00–100.30) | 96.00 (92.00–101.00) | 0.879 |
| SBP (mm Hg) | 115.67 (105.33–128.67)*§¶ | 118.50 (107.46–133.00)¶ | 122.50 (109.33–136.67)†¶ | 123.00 (114.00–137.67)† | 130.67 (118.67–142.17)† ‡* | 0.000 |
| DBP (mm Hg) | 74.33 (68.00–81.33)§¶ | 77.00 (70.00–83.75) | 76.33 (69.00–83.33)¶ | 77.50 (72.33–82.67)† | 79.00 (72.33–88.17)†* | 0.000 |
| FPG (mmol/L) | 5.08 (4.83–5.29)‡§¶ | 5.20 (4.98–5.38)† | 5.11 (4.90–5.33) | 5.16 (4.93–5.36)† | 5.16 (4.92–5.36)† | 0.000 |
| 2hPG (mmol/L) | 6.15 (5.40–6.88)*§ | 6.14 (5.45–6.93) | 6.40 (5.67–7.09)† | 6.54 (5.85–7.10)† | 6.33 (5.50–7.08) | 0.000 |
| HbA1c (%) | 5.60 (5.30–5.90)*§¶ | 5.70 (5.48–5.90) | 5.70 (5.40–5.90)† | 5.70 (5.50–6.00)† | 5.70 (5.40–6.00)† | 0.000 |
| TG (mmol/L) | 1.10 (0.80–1.60) | 1.12 (0.80–1.63) | 1.11 (0.84–1.59) | 1.14 (0.89–1.60) | 1.07 (0.81–1.50) | 0.494 |
| TC (mmol/L) | 4.46±1.01 | 4.45±1.17 | 4.50±1.02 | 4.72±1.14 | 4.52±1.10 | 0.062 |
| HDL–c (mmol/L) | 1.32 (1.09–1.60) | 1.32 (1.05–1.52) | 1.30 (1.08–1.56) | 1.36 (1.20–1.57) | 1.31 (1.09–1.60) | 0.376 |
| LDL–c (mmol/L) | 2.51 (2.04–3.03) | 2.44 (1.97–3.09) | 2.53 (1.99–2.99) | 2.65 (2.06–3.17) | 2.45 (1.95–3.01) | 0.688 |
| Family history of diabetes | 119 (12.24%) | 12 (7.55%) | 29 (8.48%) | 18 (11.84%) | 29 (11.15%) | 0.214 |
| Current smoker | 137 (14.10%) | 28 (17.61%) | 42 (12.28%) | 18 (11.84%) | 44 (16.92%) | 0.307 |
| Physical activity | 719 (73.97%) | 107 (67.30%) | 255 (74.56%) | 113 (74.34%) | 195 (75.00%) | 0.435 |

Data are expressed as mean±SD or median (IQR) or n (%).

$X^2$ test was used to compare gender compositions, family history of diabetes, current smoking status and physical activity among five groups. If needed, $X^2$ segmentation was applied for further comparisons between any two subgroups with an adjusted significance level (a'=0.005).

Kruskal-Wallis H analysis was applied for follow-up time among five groups. Mann-Whitney U analysis was performed for comparison within any two subgroups additionally (a'=0.005).

One-way ANOVA was used for the rest measurements among five groups, while least significant difference (LSD) analysis was applied for age, SBP, DBP, FPG, 2hPG and HbA1c comparisons between any two subgroups (a'=0.005).

*Versus isolated IGT and p<0.005.

†Versus NGT and p<0.005.

‡Versus isolated IFG and p<0.005.

§Versus IFG+IGT and p<0.005.

¶Versus NDDM and p<0.005.

2hPG, 2-hour plasma glucose (after oral glucose tolerance test); ANOVA, analysis of variance; DBP, diastolic blood pressure; FPG, fasting plasma glucose; HbA1c, haemoglobin A1c; HDL–c, high-density lipoprotein cholesterol; IFG, impaired fasting glucose; IGT, impaired glucose tolerance; LDL–c, low-density lipoprotein cholesterol; NDDM, newly diagnosed diabetes mellitus; NGT, normal glucose tolerance; SBP, systolic blood pressure; TC, total cholesterol; TG, triglyceride.

**Table 2** Baseline and follow-up anthropometric values in participants who developed hyperglycaemic disorders

| | NGT (n=972) | Isolated IFG (n=159) | Isolated IGT (n=342) | IFG+IGT (n=152) | NDDM (n=260) | Overall p values |
|---|---|---|---|---|---|---|
| **At follow-up survey** | | | | | | |
| WHtR (cm/cm) | 0.51 (0.47–0.55)*†‡ | 0.52 (0.48–0.56)†‡ | 0.53 (0.49–0.57)†‡§ | 0.54 (0.51–0.59)*§¶ | 0.56 (0.52–0.60)*§¶ | 0.000 |
| BMI (kg/m²) | 23.46 (21.77–25.53)*†‡¶ | 24.27 (22.49–26.17)‡§ | 24.44 (22.63–26.50)‡§ | 25.09 (23.62–27.01)§ | 25.73 (23.29–27.82)*§¶ | 0.000 |
| WC (cm) | 80.65 (74.00–87.00)*†‡¶ | 82.80 (77.00–91.00)†‡§ | 84.00 (78.00–90.00)†‡§ | 86.70 (80.28–93.00)*§¶ | 88.00 (82.00–95.00)*§¶ | 0.000 |
| WHR (cm/cm) | 0.86 (0.81–0.91)*†‡¶ | 0.88 (0.84–0.92)‡§ | 0.88 (0.83–0.92)‡§ | 0.90 (0.86–0.94)§ | 0.91 (0.87–0.95)*§¶ | 0.000 |
| **At baseline survey** | | | | | | |
| WHtR (cm/cm) | 0.50±0.05*†¶ | 0.52±0.06†‡§ | 0.53±0.05‡¶ | 0.54±0.05§¶ | 0.55±0.06*§¶ | 0.000 |
| BMI (kg/m²) | 23.03 (21.23–25.16)*†‡ | 23.31 (21.56–25.64)‡ | 24.03 (22.10–26.22)§ | 24.98 (23.47–26.67)§ | 25.42 (23.17–27.22)§¶ | 0.000 |
| WC (cm) | 79.00 (73.00–86.00) | 82.00 (76.00–89.00) | 83.00 (77.10–89.00) | 87.00 (81.00–91.28) | 86.00 (80.00–93.00) | 0.282 |
| WHR (cm/cm) | 0.86 (0.81–0.90)‡ | 0.87 (0.92–0.92) | 0.87 (0.82–0.91) | 0.89 (0.86–0.93) | 0.90 (0.86–0.94)§ | 0.010 |

Data are expressed as median (IQR) or mean±SD.

At follow-up survey: one-way ANOVA was used for WHtR, BMI and WC among the five glucose metabolic groups. LSD analysis was applied for the further comparisons between any two subgroups (a'=0.005). Kruskal-Wallis H analysis was applied for WHR among the five groups and Mann-Whitney U analysis was performed for the following comparisons within any two subgroups (a'=0.005).

At baseline survey: one-way ANOVA was used for all indices among the five glucose metabolic groups. LSD analysis was applied for WHtR, BMI and WHR between any two subgroups' comparison (a'=0.005).

*Versus isolated IGT and p<0.005.
†Versus IFG+IGT and p<0.005.
‡Versus NDDM and p<0.005.
§Versus NGT and p<0.005.
¶Versus isolated IFG and p<0.005.

ANOVA, analysis of variance; BMI, body mass index; IFG, impaired fasting glucose; IGT, impaired glucose tolerance; NDDM, newly diagnosed diabetes mellitus; NGT, normal glucose tolerance; WC, waist circumference; WHR, waist-to-hip ratio; WHtR, waist-to-height ratio.

when all the subjects were still normoglycaemic. Baseline WHtR, BMI and WHR, but not WC, substantially differed among the five groups (p<0.05) (table 2). NGT subjects had lower WHtR than the subjects who subsequently developed hyperglycaemia (p<0.005). The WHtR values of the IFG+IGT and NDDM groups were higher than those of the isolated IFG group (p<0.005), while the isolated IGT group had a lower WHtR than the NDDM group (p<0.005). The BMI of the NGT group was lower than those of the isolated IGT, IFG+IGT and NDDM groups (p<0.005), and the isolated IFG group had a lower BMI than NDDM subjects (p<0.005). In addition, NGT individuals had a lower WHR than patients with NDDM at baseline (p<0.005). Consistent with the findings at follow-up, it is worth noting that at baseline, there were no significant differences in WHtR, BMI and WHR between individuals who subsequently developed IFG+IGT and those who converted to NDDM (p>0.005).

### Use of baseline anthropometric indices to predict future prediabetes and NDDM

For the prediction of isolated IFG, baseline WHtR, WC and WHR showed significantly different areas under the curve (AUCs) (p<0.05) (table 3). WHtR and WC were more effective at predicting isolated IFG than BMI (p<0.05) (figure 1A). For subjects who developed isolated IGT, the AUCs of all the four indices were significant (p=0.000). WHtR had a higher predictive value than BMI, WC and WHR (p<0.05), while WC was superior to WHR for predicting isolated IGT (p<0.05) (figure 1B). For IFG+IGT incidence, all four parameters were valuable predictors (p=0.000), among which WHtR and WC ranked higher than WHR (p<0.05) (figure 1C). For the prediction of NDDM, the four indices were significant (p<0.05), but WHtR was the best predictor (p<0.05) (figure 1D). The optimal thresholds for predicting hyperglycaemia for the four indices (WC and WHR thresholds for men and women) were then calculated.

### Multivariate analysis of baseline anthropometric indices with respect to risk of subsequent prediabetes and NDDM

According to Cox proportional hazards regression, the risk of developing isolated IFG was greater with higher WC at baseline (p<0.05) (table 4). The risk factors for the development of isolated IGT were baseline WHtR, BMI and WC (p<0.05). For both IFG+IGT and NDDM, high baseline WHtR, BMI, WC and WHR were all risk factors (p<0.05).

**Table 3** ROC curve analysis of baseline anthropometric indices for predicting future hyperglycaemia

| | AUC | SE | P values | 95% CI | Cut-off point | Youden's value | Sensitivity % | Specificity % | DeLong's test (p values) | | | |
| --- | --- | --- | --- | --- | --- | --- | --- | --- | --- | --- | --- | --- |
| | | | | | | | | | WHtR (cm/cm) | BMI (kg/m²) | Waist circumference (cm) | WHR (cm/cm) |
| **Isolated IFG** | | | | | | | | | | | | |
| WHtR (cm/cm) | 0.578 | 0.025 | 0.002 | (0.529 to 0.626) | 0.51 | 0.151 | 54.90 | 60.19 | – | 0.010 | 0.201 | 0.611 |
| BMI (kg/m²) | 0.544 | 0.025 | 0.081 | (0.495 to 0.593) | 21.36 | 0.078 | 80.40 | 27.43 | 0.010 | – | 0.023 | 0.421 |
| Waist circumference (cm) | 0.592 | 0.024 | 0.000 | (0.545 to 0.639) | 77.10 | 0.148 | 71.24 | 43.54 | 0.201 | 0.023 | – | 0.195 |
| Women | 0.584 | 0.031 | 0.010 | (0.524 to 0.644) | 75.00 | 0.166 | 74.44 | 42.11 | – | – | – | – |
| Men | 0.579 | 0.041 | 0.050 | (0.526 to 0.631) | 87.00 | 0.165 | 49.21 | 67.24 | – | – | – | – |
| WHR (cm/cm) | 0.567 | 0.026 | 0.008 | (0.537 to 0.597) | 0.88 | 0.128 | 47.06 | 65.71 | 0.611 | 0.421 | 0.195 | – |
| Women | 0.568 | 0.033 | 0.036 | (0.504 to 0.632) | 0.85 | 0.140 | 57.78 | 56.19 | – | – | – | – |
| Men | 0.525 | 0.042 | 0.534 | (0.471 to 0.578) | 0.90 | 0.095 | 53.97 | 55.52 | – | – | – | – |
| **Isolated IGT** | | | | | | | | | | | | |
| WHtR (cm/cm) | 0.634 | 0.017 | 0.000 | (0.600 to 0.667) | 0.51 | 0.214 | 62.24 | 59.12 | – | 0.003 | 0.006 | 0.000 |
| BMI (kg/m²) | 0.591 | 0.018 | 0.000 | (0.556 to 0.627) | 22.68 | 0.155 | 68.88 | 46.64 | 0.003 | – | 0.178 | 0.223 |
| Waist circumference (cm) | 0.610 | 0.017 | 0.000 | (0.576 to 0.645) | 78.00 | 0.197 | 71.90 | 47.81 | 0.006 | 0.178 | – | 0.001 |
| Women | 0.635 | 0.021 | 0.000 | (0.593 to 0.676) | 78.00 | 0.260 | 68.42 | 57.59 | – | – | – | – |
| Men | 0.542 | 0.032 | 0.174 | (0.480 to 0.605) | 87.80 | 0.132 | 43.44 | 69.76 | – | – | – | – |
| WHR (cm/cm) | 0.567 | 0.018 | 0.000 | (0.539 to 0.594) | 0.86 | 0.123 | 61.63 | 50.64 | 0.000 | 0.223 | 0.001 | – |
| Women | 0.587 | 0.022 | 0.000 | (0.544 to 0.630) | 0.82 | 0.154 | 77.03 | 38.39 | – | – | – | – |
| Men | 0.524 | 0.032 | 0.433 | (0.463 to 0.586) | 0.89 | 0.098 | 57.38 | 52.41 | – | – | – | – |
| **IFG+IGT** | | | | | | | | | | | | |
| WHtR (cm/cm) | 0.713 | 0.022 | 0.000 | (0.670 to 0.755) | 0.53 | 0.351 | 62.33 | 72.79 | – | 0.106 | 0.556 | 0.026 |
| BMI (kg/m²) | 0.685 | 0.022 | 0.000 | (0.642 to 0.729) | 23.38 | 0.316 | 77.40 | 54.22 | 0.106 | – | 0.254 | 0.492 |
| Waist circumference (cm) | 0.706 | 0.021 | 0.000 | (0.665 to 0.748) | 79.80 | 0.351 | 82.88 | 52.19 | 0.556 | 0.254 | – | 0.032 |
| Women | 0.732 | 0.026 | 0.000 | (0.682 to 0.783) | 79.80 | 0.420 | 79.57 | 62.38 | – | – | – | – |
| Men | 0.656 | 0.039 | 0.000 | (0.579 to 0.733) | 90.30 | 0.242 | 43.40 | 80.76 | – | – | – | – |
| WHR (cm/cm) | 0.667 | 0.022 | 0.000 | (0.638 to 0.695) | 0.87 | 0.274 | 69.18 | 58.23 | 0.026 | 0.492 | 0.032 | – |
| Women | 0.686 | 0.027 | 0.000 | (0.633 to 0.739) | 0.83 | 0.312 | 86.02 | 45.20 | – | – | – | – |
| Men | 0.631 | 0.038 | 0.003 | (0.556 to 0.705) | 0.92 | 0.261 | 54.72 | 71.38 | – | – | – | – |

Continued

**Table 3** Continued

| | AUC | SE | P values | 95% CI | Cut-off point | Youden's value | Sensitivity, % | Specificity, % | DeLong's test (p values) | | | |
| --- | --- | --- | --- | --- | --- | --- | --- | --- | --- | --- | --- | --- |
| | | | | | | | | | WHtR (cm/cm) | BMI (kg/m²) | Waist circumference (cm) | WHR (cm/cm) |
| **NDDM** | | | | | | | | | | | | |
| WHtR (cm/cm) | 0.730 | 0.017 | 0.000 | (0.696 to 0.764) | 0.52 | 0.366 | 74.21 | 62.43 | – | 0.000 | 0.001 | 0.010 |
| BMI (kg/m²) | 0.677 | 0.020 | 0.000 | (0.639 to 0.716) | 24.32 | 0.315 | 64.68 | 66.81 | 0.000 | – | 0.093 | 0.596 |
| Waist circumference (cm) | 0.700 | 0.018 | 0.000 | (0.665 to 0.735) | 78.00 | 0.292 | 81.35 | 47.81 | 0.001 | 0.093 | – | 0.429 |
| Women | 0.714 | 0.021 | 0.000 | (0.673 to 0.756) | 77.10 | 0.344 | 81.76 | 52.63 | – | – | – | – |
| Men | 0.686 | 0.033 | 0.000 | (0.622 to 0.750) | 88.00 | 0.298 | 56.99 | 72.85 | – | – | – | – |
| WHR (cm/cm) | 0.688 | 0.018 | 0.000 | (0.661 to 0.715) | 0.88 | 0.304 | 67.73 | 62.71 | 0.010 | 0.596 | 0.429 | – |
| Women | 0.696 | 0.022 | 0.000 | (0.653 to 0.738) | 0.84 | 0.301 | 79.75 | 50.31 | – | – | – | – |
| Men | 0.681 | 0.030 | 0.000 | (0.622 to 0.740) | 0.92 | 0.299 | 60.22 | 69.66 | – | – | – | – |

AUC, area under curve; BMI, body mass index; IFG, impaired fasting glucose; IGT, impaired glucose tolerance; NDDM, newly diagnosed diabetes mellitus; ROC, receiver operating characteristic; WHR, waist-to-hip ratio; WHtR, waist-to-height ratio.

## DISCUSSION

In this community-based prospective cohort study, we have shown that: (1) for patients with hyperglycaemia, WHtR, BMI, WC and WHR tended to be as follows: NGT < isolated IFG and isolated IGT < IFG+IGT and NDDM; (2) among these categories of hyperglycaemia, it is noteworthy that unlike isolated IFG and isolated IGT, there were no significant differences in baseline WHtR or BMI between subjects with IFG+IGT and NDDM; (3) WHtR, BMI, WC and WHR could predict the presence of prediabetes or diabetes 3 years in advance; furthermore, the greater were these baseline anthropometric values, the higher was the risk of developing hyperglycaemia; (4) optimal threshold values for the four variables for identification of prediabetes and diabetes were calculated, with WHtR performing best of these in the prediction of hyperglycaemia.

An Iranian study of 5879 people 9 years after they were initially found to be normoglycaemic found that 1755 subjects had developed prediabetes and that isolated IFG was the most common prediabetic phenotype. This study found that among women, in contrast to the use of BMI, hip circumference and WC, WHtR was the only significant anthropometric predictor of prediabetes.[14] Lyssenko et al reported a study of 1190 subjects in Finland who initially had NGT. During a median follow-up of 6 years, 199 had progressed to prediabetes. Compared with those who remained NGT, those with prediabetes had substantially higher BMI and WHtR at baseline.[15] Many investigators have shown that anthropometry is tightly correlated with the occurrence of prediabetes although most of the studies conducted have been cross-sectional, rather than longitudinal.[16–19]

After reviewing the literature, we found some common themes: (1) with respect to prediabetes, the majority of the studies only defined one or two distinct prediabetic phenotypes or defined a single category called 'prediabetes'; (2) rarely did investigators describe the respective anthropometric characteristics of the various hyperglycaemic disorders in their manuscripts. We located only one previous report that gave anthropometric information in detail for all the potential prediabetic phenotypes and NDDM.[20] It was shown in this study that WHtR, BMI, WC and WHR varied substantially among subjects with NGT, isolated IFG, isolated IGT, IFG+IGT and NDDM, but none of the anthropometric indices were compared between hyperglycaemic groups. Therefore, the possibility that anthropometry might vary between prediabetes and NDDM could not be assessed, and moreover, this study was cross-sectional. To our knowledge, the present work is the first prospective cohort study that described the anthropometric characteristics of participants who progressed to diverse hyperglycaemic conditions and demonstrated the variation among WHtR, BMI, WC and WHR in the transition from NGT to prediabetes and overt NDDM.

The pathogenesis of isolated IFG and isolated IGT is heterogeneous, while individuals with IFG+IGT

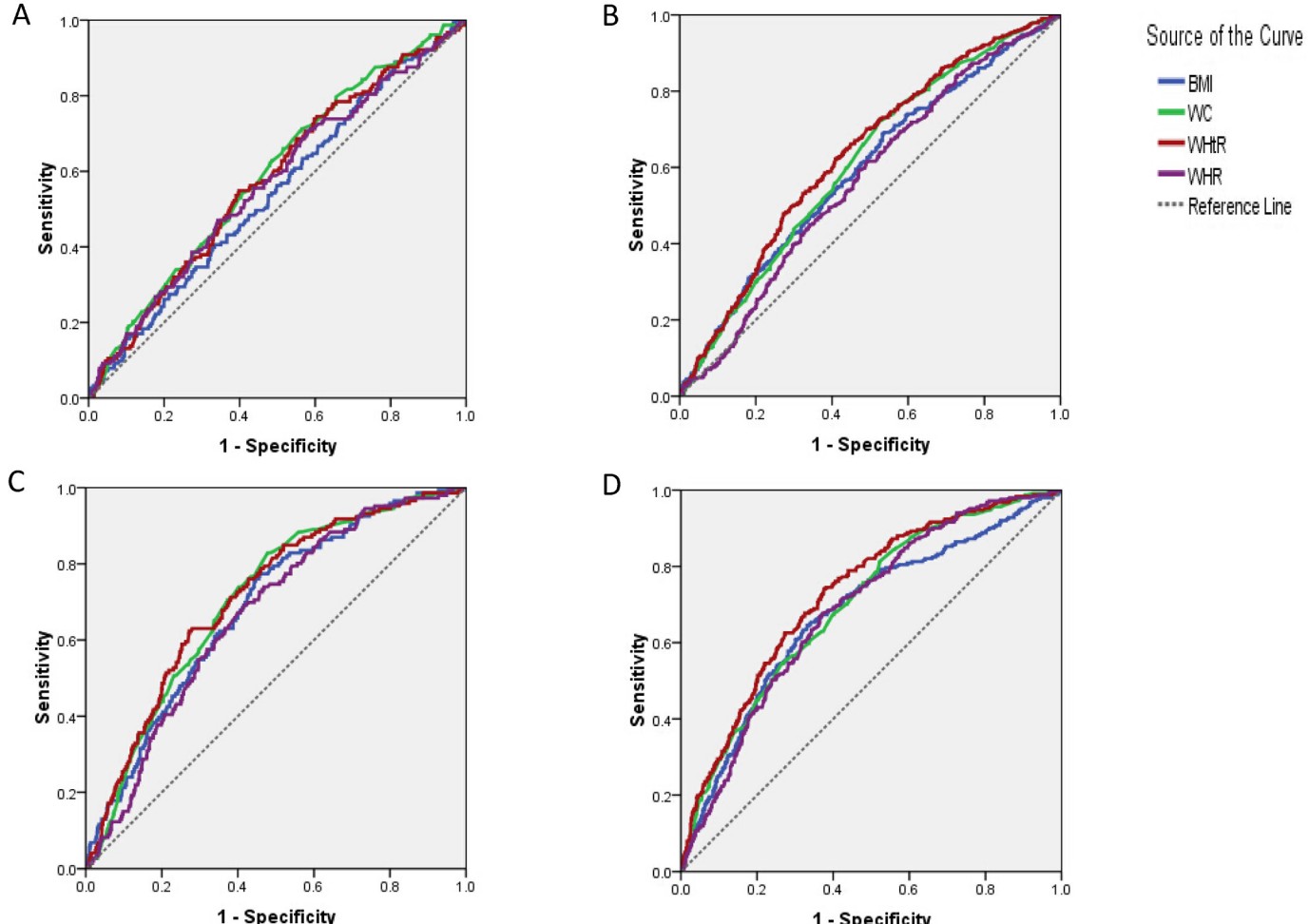

**Figure 1** ROC curves of baseline anthropometric indices in subjects who developed (A) isolated IFG, (B) isolated IGT, (C) IFG+IGT and (D) NDDM. BMI, body mass index; IFG, impaired fasting glucose; IGT, impaired glucose tolerance; NDDM, newly diagnosed diabetes mellitus; ROC, receiver operating characteristic; WC, waist circumference; WHR, waist-to-hip ratio; WHtR, waist-to-height ratio.

manifest both hepatic and peripheral insulin resistance. Prediabetes, as an intermediate hyperglycaemic state, carries a high risk for the subsequent development of diabetes. Among the three prediabetic phenotypes, IFG+IGT carries approximately twice the risk of transition to diabetes compared with subjects with just one of the abnormalities.[21] In our previous work, we found that several biomarkers in individuals with IFG+IGT had similar values to those present in the NDDM population, but these were different in individuals with IFG or IGT alone.[22–24] Consistent with this, in the present study, we observed that participants who subsequently developed hyperglycaemia had higher WHtR, BMI and WHR at baseline than those who remained NGT. Among the three prediabetic phenotypes, IFG+IGT subjects had the most adverse anthropometric profiles at baseline, such that there were no significant differences from the NDDM group. These findings may imply that although IFG+IGT is a subtype of prediabetes, some aspects of its pathophysiology have already deteriorated to the same extent as in NDDM. However, prediabetes is a reversible condition

and consequently, prompt intervention is required to avoid or delay its progression, especially for patients with IFG+IGT.

A prospective study conducted in Pima Indians found that BMI and WHtR were the best predictors of diabetes in men, while BMI, WHtR, WC and waist-to-thigh ratio were the best predictors in women.[25] Chei *et al* published a cohort study of 5617 Japanese participants, finding that in women only, the significant predictors of T2D were BMI, WC and WHtR.[26] Finally, in a multiethnic cohort of 1073 non-Hispanic white, Hispanic and African-American non-diabetic individuals, baseline anthropometric information showed that BMI was most predictive of diabetes in the non-Hispanic white and Hispanic populations, whereas all the indicators of central obesity were more predictive than measures of overall adiposity in the African-American population.[27] The contrasts in these sets of data indicate that the validity of such anthropometric measurements for the prediction of diabetes development vary among different ethnicities, genders and regions. Based on our ROC analysis, WHtR was most effective for

**Table 4** Multivariate analysis of baseline anthropometric indices with respect to risk of subsequent prediabetes and NDDM

| | Isolated IFG | | Isolated IGT | | IFG+IGT | | NDDM | |
|---|---|---|---|---|---|---|---|---|
| | HR (95% CI) | P values | HR (95% CI) | P values | HR (95% CI) | P values | HR (95% CI) | P values |
| WHtR (cm/cm) | 1.471 (0.901 to 2.402) | 0.123 | 1.951 (1.550 to 2.457) | 0.000 | 3.002 (2.137 to 4.216) | 0.000 | 2.765 (2.065 to 3.703) | 0.000 |
| BMI (kg/m²) | 1.186 (0.699 to 2.012) | 0.526 | 1.571 (1.241 to 1.988) | 0.000 | 3.298 (2.224 to 4.892) | 0.000 | 2.305 (1.773 to 2.998) | 0.000 |
| WC (cm) | 1.603 (1.112 to 2.310) | 0.011 | 1.644 (1.275 to 2.118) | 0.000 | 4.570 (2.948 to 7.084) | 0.000 | 2.666 (1.886 to 3.769) | 0.000 |
| WHR (cm/cm) | 1.182 (0.739 to 1.889) | 0.486 | 0.972 (0.724 to 1.304) | 0.848 | 1.571 (1.003 to 2.465) | 0.048 | 1.706 (1.196 to 2.433) | 0.003 |

Cox proportional hazards models were used to calculate HR and 95% CI. A univariate analysis was performed for each potential risk factor first, including age (years), gender (male/female), systolic blood pressure (mm Hg), diastolic blood pressure (mm Hg), fasting plasma glucose (mmol/L), 2-hour plasma glucose (mmol/L) (after oral glucose tolerance test), HbA1c (%), total cholesterol (mmol/L), triglyceride (mmol/L), high-density lipoprotein cholesterol (mmol/L), low-density lipoprotein cholesterol (mmol/L), diabetes family history (yes/no), current smoking status (yes/no), physical activity situation (yes/no), WHtR (low/high), BMI (low/high), WC (low/high) and WHR (low/high). The four anthropometric indicators were dichotomised into low or high level by using cut-off values derived from previous ROC curve analysis. Then those risk factors with a p value <0.2 in univariate analysis were selected to enter the multivariate model.
BMI, body mass index; HbA1c, haemoglobin A1c; IFG, impaired fasting glucose; IGT, impaired glucose tolerance; NDDM, newly diagnosed diabetes mellitus; ROC, receiver operating characteristic; WC, waist circumference; WHR, waist-to-hip ratio; WHtR, waist-to-height ratio.

the prediction of prediabetes and overt NDDM, followed by WC, while BMI and WHR were relatively weak predictors. Results from two western Pacific studies were consistent with our findings.[28 29]

A systematic review proposed that the threshold values for WHtR in the prediction of diabetes in men and women are 0.52 and 0.53, respectively.[30] In a Chinese community-based prospective cohort study, the optimal threshold values for WHtR and BMI were 0.51 and 24 for men, and 0.55 and 25 for women, respectively.[29] These predictive values were similar to those identified in our study.

Several limitations to our work should be addressed. First, the follow-up period of a median 3.00 years was relatively short. However, we identified high cumulative incidences of prediabetes and NDDM (34.6% and 13.8%, respectively). The fast pace of life and sedentary lifestyle of the population may be the main contributors to the rapid growth in hyperglycaemia. However, it might also be the result of selection bias because subjects with a higher risk might be more likely to take part in the follow-up assessment. In addition, the participants were ≥40 years old, a little older than the subjects (≥35 years) in some other epidemiological studies. This might also be an explanation that a large proportion of subjects became hyperglycaemic in this cohort study. Second, the proportion of participants attending the follow-up assessment was low (41.91%). Conducting of a phone interview once a year at least, followed by prompt examination, could improve this statistic in the future. Third, the sample size was limited. On account of this weakness, it was not possible to calculate anthropometric threshold values for each hyperglycaemic state by gender. Further studies are required to establish specific screening thresholds for prediabetes and NDDM in men and women, especially with regard to WC and WHR. Fourth, there was lack of OGTT reproducibility in each set of measurements. Unwillingness of subjects and limited staff and financial resources were the two major causes of this. By combining these data with the questionnaire data and the HbA1c results, we tried to minimise the associated error and improve the diagnostic accuracy as much as possible.

In summary, WHtR, BMI, WC and WHR are all predictors of the development of prediabetes and NDDM 3 years in advance. Individuals with high WHtR, BMI, WC and WHR are thus at higher risk of developing prediabetes and T2D. The optimal thresholds for all the anthropometric measures to predict hyperglycaemia were calculated, with a WHtR value of 0.52 performing best at predicting the development of isolated IFG or IGT, IFG+IGT and NDDM. The magnitude of WHtR and BMI in normoglycaemic subjects illustrate the likelihood of progression from normoglycaemia to prediabetes, and then to overt T2D. Of note, and in contrast to the situation with regard to isolated IFG or IGT, the anthropometric characteristics of IFG+IGT subjects were similar to those of the NDDM population, both at baseline and follow-up.

**Author affiliations**
¹Division of Endocrinology and Metabolism, West China Hospital of Sichuan University, Chengdu, China
²Division of Endocrinology and Metabolism, Hospital Affiliated to Southwest Medical University, Luzhou, China
³Division of Endocrinology and Metabolism, The Fifth People's Hospital of Chengdu, Chengdu, China
⁴Chengdu Yinchao Community Hospital, Chengdu, China

**Acknowledgements** All the authors appreciated the doctors and other staff from primary healthcare centres for health report suggestions, materials supplement and participant organisation.

**Contributors** All the authors engaged in the surveys. FZ and NT designed this article. QW, HC, DL and QY acquired and collected the data. JL, ZY, QL and YZ organised all the data. FZ, QW and HC analysed all the information. FZ and LT drafted the manuscript. FZ and NT revised the article critically. All the authors read and approved the final manuscript.

**Funding** The surveys in Luzhou were supported by the grants from the National Clinical Research Center for Metabolic Diseases (2013BAI09B13), the National Key New Drug Creation and Manufacturing Program of Ministry of Science and Technology (2012ZX09303006—001). The screenings in Wenjiang were supported by the National Key Technology R&D Program of China (Grant No. 2009BAI80B01 and No. 2009BAI80B02) and Sichuan Provincial Science and Technology Foundation (Grant No. 150022 and No. 2015SZ0228).

**Competing interests** None declared.

**Patient consent** Obtained.

**Ethics approval** The Medical Ethics Committee of Hospital affiliated to Southwest Medical University, the Committee on Human Research at the Fifth People's Hospital of Chengdu.

**Provenance and peer review** Not commissioned; externally peer reviewed.

**Data sharing statement** A supplementary profile will be available online which contains comprehensive figure and table of used input data. Inquiries about additional unpublished data could be contacted with the corresponding author.

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
