## [Reviewer comments · BMJ Open]

ARTICLE DETAILS

TITLE (PROVISIONAL)	Identical anthropometric characteristics of impaired fasting glucose combined with impaired glucose tolerance and newly-diagnosed type 2 diabetes: anthropometric indicators to predict hyperglycemia in a community-based prospective cohort study in southwest China
AUTHORS	Zhang, Fang; Wan, Qin; Cao, Hongyi; Tang, Lizhi; Li, Daigang; Lü, Qingguo; Yan, Zhe; Li, Jing; Yang, Qiu; Zhang, Yuwei; Tong, Nanwei

VERSION 1 – REVIEW

REVIEWER	Dae Jung Kim Ajou University School of Medicine, Korea
REVIEW RETURNED	06-Oct-2017

GENERAL COMMENTS	This study was aimed to assess the association between the anthropometric measurement and development of hyperglycemia. Main finding was that waist height ratio is the best indicator to predict hyperglycemia. 1. I don't know why the authors divided prediabetes into isolated IFG or IGT. In the table 1, I cannot find any difference of clinical characteristics, except age, between isolated IFG and isolated IGT.2. In the conclusion, the author described that WHtR is best indicator. However, Waist circumference might be comparable, and HR is highest in isolated IFG or combined IFG/IGT group.3. I think isolated IFG is more common than isolated IGT. But in this data, isolated IFG is very small. What's your opinion?4. There is a limitation of low reproducibility of the oral GTT. You can mention it in the discussion.5. In your data, annual incidence is about 4.6% in diabetes and 11.5% in prediabetes. Although the authors mentioned lifestyle changes, is this rapid change possible in a short time? It might be related to selection bias that higher risk subjects were followed up in the next examination.
--

REVIEWER	Dr. Bishwajit Bhowmik Centre for Global Health Research Diabetic Association of Bangladesh Dhaka, Bangladesh
REVIEW RETURNED	16-Nov-2017

GENERAL COMMENTS	Dear Editor Journal of Diabetes Thank you for giving opportunity to review the manuscript ‘identical anthropometry characteristics of impaired fasting glucose combined with impaired glucose tolerance and newly-diagnosed type 2 diabetes: anthropometric indicators for hyperglycemia prediction in a community-based prospective cohort study’. Please find below mentioned comments. Sincerely yours Dr Bishwajit Bhowmik  Reviewer comments Thank you for submitting the valuable article to Journal of Diabetes. I have appreciated to give a chance for reviewing your article. The article “Identical anthropometry characteristics of impaired fasting glucose combined with impaired glucose tolerance and newly-diagnosed type 2 diabetes: anthropometric indicators for hyperglycemia prediction in a community-based prospective cohort study” is an informative article which has assessed the anthropometry characteristics in euglycemic individuals who developed hyperglycemia subsequently and to evaluate the validity for pre-diabetes and diabetes identification by anthropometric indices in Southwest China. I presume that the current study provides clinically valuable information, however the authors should reply to some questions and criticism. Major Observation: Comment 1: Author has described the article as a community-based prospective cohort study. Reviewer has gone through the method section and also the flow chart. Have you followed the same participants in all the time points or have some participants entered in time beings? If it is happened, then, how do you call it cohort? Please clarify it. Minor observation: Abstract: Comment 1: In Participants and setting section, author has mentioned Pre-diabetes-free and diabetes-free residents. Author
---

	can simple write normoglycemic participants. Introduction: Comment: 1. Introduction needs few more organization and coherence Materials and Methods: Comment 1: Author has recruited participants aged of 40-89, when good number studies in Asian countries have been shown younger age onset of diabetes and prediabetes. Please clarify it. Comment 2: Lipid profiles and physical activity have not been defined as they have been shown in the table 1. Comment 3: Author should add few more lines on laboratory procedures. Results: Comment 1. it is better to write the age of year without decimal. Comment 2. Characteristics of all subjects at baseline in supplemental table 1 should be described clearly Comment 3. Symbols at the baseline survey in table 2 should be checked, especially with the last line of the foot note. Discussion: Comment: 1. There is a spelling error (wrist) which need to be rectified in the second para of the discussion part. 2. It is understandable that in para four, increased WC should be added with the other parameters for development of hyperglycemia in future. Conclusions: The manuscript cannot be recommended for publication as it stands now but it could be an interesting manuscript for publication if they considered the aforementioned comments.
--	--

VERSION 1 – AUTHOR RESPONSE

Response to editor and reviewers

First of all, we would like to thank both the editor and reviewers for their positive and constructive comments and suggestions. Our responses to those comments are as following:

Editor:

- Please include the country in the title.

Authors (A): Dear editor, we have added the country in the title.

- The Strengths and Limitations section should just consist of point on the strengths and limitations of the study and study design. It should not serve as an article summary.

A: Dear editor, we have revised the Strengths and Limitations section according to your comments.

- Please ensure that your manuscript is proofread by a native English speaker prior to resubmission.

A: Dear editor, we improved the whole scientific English language of this manuscript and employed a professional English-editing service to polish our wording. We hope the manuscript in this version is much easier to follow and understand. Thanks for your suggestions.

Reviewer #1:

Reviewer Name: Dae Jung Kim

Institution and Country: Ajou University School of Medicine, Korea

Comments for the authors:

This study was aimed to assess the association between the anthropometric measurement and development of hyperglycemia. Main finding was that waist height ratio is the best indicator to predict hyperglycemia.

A: Dear Dr. Kim, thank you very much for your constructive advice. Our point-by-point responses to your comments are listed as below.

1. I don't know why the authors divided prediabetes into isolated IFG or IGT. In the table 1, I cannot find any difference of clinical characteristics, except age, between isolated IFG and isolated IGT.

A: Dear reviewer, thanks for your comment. Because based on the levels of fasting plasma glucose and 2-hour plasma glucose after OGTT, pre-diabetes is categorized into three different phenotypes which are isolated IFG, isolated IGT, and IFG combined with IGT. Therefore, we divided pre-diabetes into the three groups according to its classification, instead of the clinical characteristics of participants.

2. In the conclusion, the author described that WHtR is best indicator. However, Waist circumference might be comparable, and HR is highest in isolated IFG or combined IFG/IGT group.

A: Dear reviewer, as you said, waist circumference is comparable to WHtR in the prediction of hyperglycemia for most of the time. In some cases, waist circumference is even superior to WHtR. However, these conclusions were only drawn from the COX regression analysis. In the ROC curve analyses, it was found that WHtR was the best predictor for hyperglycemia, including isolated IFG or IGT, IFG+IGT, and NDDM. Overall, after combining all the results, WHtR is considered to be the best indicator for future hyperglycemia prediction, followed by waist circumference.

3. I think isolated IFG is more common than isolated IGT. But in this data, isolated IFG is very small. What's your opinion?

A: Dear reviewer, thanks for your comment. The most possible reason why isolated IFG less common than isolated IGT in this study was that the majority of Chinese pre-diabetic populations are isolated IGT patients. A national cross-sectional study in 2008 found that the incidence of isolated IGT were nearly five and four times as high as those of isolated IFG in women and men, respectively (Yang W, et al. Prevalence of Diabetes among Men and Women in China. N Engl J Med 2010; 362: 1090-101). Another cross-sectional study conducted in Chinese, Malays, and Asian-Indians participants also demonstrated that the isolated IGT patients were four times more than the isolated IFG individuals (Alperet DJ, et al. Optimal anthropometric measures and thresholds to identify undiagnosed type 2 diabetes in three major Asian ethnic groups. Obesity (Silver Spring) 2016; 24 (10):2185-93). In addition, polished rice and refined wheat are the staple foods for most residents in southwest China, which are of high glycemic index and glucose load values. The dietary habit may contribute to the high prevalence of isolated IGT in the area.

During the progression from normoglycemia to overt type 2 diabetes, isolated IGT always emerges earlier than isolated IFG. Because we did not assess the insulin secretion, or hepatic and muscle insulin sensitivities, so we could not clarify the phenomenon from a view of pathogenetic mechanism in this work. Further prospective cohort studies are needed to explore the metabolic features in subjects with distinct hyperglycemic states in southwest China.

4. There is a limitation of low reproducibility of the oral GTT. You can mention it in the discussion.

A: Dear reviewer, thanks for your suggestion. We have mentioned the lack of OGTT reproducibility as a limitation in the seventh paragraph of the discussion part.

5. In your data, annual incidence is about 4.6% in diabetes and 11.5% in prediabetes. Although the authors mentioned lifestyle changes, is this rapid change possible in a short time? It might be related to selection bias that higher risk subjects were followed up in the next examination.

A: Dear reviewer, thanks for your kind reminder. Although we knew that the lifestyle change was not sufficient to explain the phenomenon of such high annual incidences of hyperglycemia, we did not broaden our minds to the selection bias. As you mentioned, compared with the non-risk participants, the diabetes-risk subjects might come back to the follow-up examinations more commonly, which contributes to the high cumulative incidences. Moreover, the participants we recruited were ≥ 40 years old, who were older than the subjects aged of ≥ 35 years in some other epidemiological studies. It might be another reason why there were a large proportion of hyperglycemia incidences in this cohort study. We have added these explanations in the seventh paragraph of the discussion part.

A: Dear Dr. Kim, at last, we sincerely appreciated your time and efforts for so many constructive and professional comments.

Reviewer #2:

Reviewer Name: Dr. Bishwajit Bhowmik

Institution and Country: Centre for Global Health Research, Diabetic Association of Bangladesh, Dhaka, Bangladesh

Comments for the authors:

Thank you for submitting the valuable article to Journal of Diabetes. I have appreciated to give a chance for reviewing your article. The article "Identical anthropometry characteristics of impaired fasting glucose combined with impaired glucose tolerance and newly-diagnosed type 2 diabetes: anthropometric indicators for hyperglycemia prediction in a community-based prospective cohort study" is an informative article which has assessed the anthropometry characteristics in euglycemic

individuals who developed hyperglycemia subsequently and to evaluate the validity for pre-diabetes and diabetes identification by anthropometric indices in Southwest China. I presume that the current study provides clinically valuable information, however the authors should reply to some questions and criticism.

A: Dear Dr. Bhowmik, thanks very much for your affirmation to our work. We all appreciated your time and effort for your constructive comments and suggestions. Our point-by-point responses to your comments are listed as below.

Major Observation:

Comment 1: Author has described the article as a community-based prospective cohort study. Reviewer has gone through the method section and also the flow chart. Have you followed the same participants in all the time points or have some participants entered in time beings? If it is happened, then, how do you call it cohort? Please clarify it.

A: Dear reviewer, we are sorry for the misunderstanding caused by the scientific English and flow chart we used. All the participants we followed had been screened at the beginning, who were from either the 10007 subjects in Luzhou or the 1104 residents in Wenjiang at baseline, and came from the same communities. Therefore, we assumed the study design as a community-based prospective cohort study. In order to avoid any more misleading, we polished our English writing and improved the integrity and logicity of the flow chart (Supplemental Figure 1). We hope this version is much better understandable. Thanks for your comment.

Minor observation:

Abstract:

Comment 1: In Participants and setting section, author has mentioned Pre-diabetes-free and diabetesfreeresidents. Author can simple write normoglycemic participants.

A: Dear reviewer, we have changed "Pre-diabetes-free and diabetes-free residents" into "residents with euglycemia" in the abstract.

Introduction:

Comment: 1. Introduction needs few more organization and coherence.

A: Dear reviewer, according to your suggestion, we have organized the introduction to be more coherent.

Materials and Methods:

Comment 1: Author has recruited participants aged of 40-89, when good number studies in Asiancountries have been shown younger age onset of diabetes and prediabetes. Please clarify it.

A: Dear reviewer, thanks for your comment. The participants in our work were recruited from two community-based cohort studies. The one in Luzhou was part of the Risk Evaluation of cAncers in Chinese diabeTic Individuals: a lONgitudinal (REACTION) study, which aimed to evaluate the risk of cancers in diabetic patients. Thus, the age range of all the subjects was set at 40 years and older. The study in Wenjiang was a survey for screening risk of metabolic syndrome, whose participants were also required at least 40 years old. Since both of the studies were not initially designed for screening the incidence of hyperglycemia, therefore, when compared with the studies focusing on assessing hyperglycemia prevalence among adults (aged ≥ 18 years old), the results of this work presented an older age onset of pre-diabetes and diabetes. Though we did not include the adults aged of 18—39, the participants in our studies were more convincingly diagnosed type 2 diabetes.

Comment 2: Lipid profiles and physical activity have not been defined as they have been shown in the table 1.

A: Dear reviewer, we have added the definitions of lipid profile and physical activity in the Paragraph Seven and Six in the methods part, respectively.

Comment 3: Author should add few more lines on laboratory procedures.

A: Dear reviewer, we have added more information on the laboratory procedures in the methods part.

Results:

Comment 1. it is better to write the age of year without decimal.

A: Dear reviewer, we have revised the format of age expression according to your suggestion.

Comment 2. Characteristics of all subjects at baseline in supplemental table 1 should be described clearly.

A: Dear reviewer, we have added some descriptions on Supplemental Table 1 and Table 1 in the first paragraph of the results part.

Comment 3. Symbols at the baseline survey in table 2 should be checked, especially with the last line of the foot note.

A: Dear reviewer, we have improved the foot note and symbols of Table 2, and added some descriptions in the relevant text in the third paragraph of the results part. We believe it is easier to follow now. Thanks for your comment.

Discussion:

Comment: 1. There is a spelling error (wrist) which needs to be rectified in the second paragraph of the discussion part.

A: Dear reviewer, we have corrected the spelling error to word "waist" in this paragraph. Thanks for your meticulousness.

2. It is understandable that in paragraph four, increased WC should be added with the other parameters for development of hyperglycemia in future.

A: Dear reviewer, thanks for your suggestion. As you mentioned, increased WC at baseline was a potential risk of hyperglycemia. However, when we assessed the anthropometric characteristics of participants at baseline, WC did not show a significant difference between the subjects who developed hyperglycemia and those who still remained normoglycemic in the future. Therefore, WC has not been added with WHtR, BMI, or WHR in the Paragraph Four, where only the latter three parameters at baseline were demonstrated significantly different between the subsequent hyperglycemic and euglycemic subjects.

Conclusions: The manuscript cannot be recommended for publication as it stands now but it could be an interesting manuscript for publication if they considered the aforementioned comments.

A: Dear Dr. Bhowmik, we have carefully considered your comments and revised our manuscript according to the suggestions. At last, we sincerely appreciated your time and efforts for so many constructive and meticulous comments.

VERSION 2 – REVIEW

REVIEWER	Dae Jung Kim Ajou University School of Medicine, South Korea
REVIEW RETURNED	13-Dec-2017

GENERAL COMMENTS	This manuscript was well revised according to the reviewers' comments. I think it is good for publication as it is. But I still have a question about the prevalence of isolated IFG in your cohorts. I mentioned that isolated IGT is more common than isolated IFG in two good studies. However, there is difference of definition of IFG. In two studies you mentioned, IFG was defined as FPG of 110-125mg/dL. So I can agree that isolated IFG is less than isolated IGT. In your studies, IFG was defined as FPG of 100-125mg/dL. I can guess isolated IFG is more common than isolated IGT.
---

VERSION 2 – AUTHOR RESPONSE

Response to reviewer:

Reviewer #1:

Reviewer Name: Dae Jung Kim

Institution and Country: Ajou University School of Medicine, Korea

Comments for the authors:

This manuscript was well revised according to the reviewers' comments.

I think it is good for publication as it is.

But I still have a question about the prevalence of isolated IFG in your cohorts.

I mentioned that isolated IGT is more common than isolated IFG in two good studies.

However, there is difference of definition of IFG.

In two studies you mentioned, IFG was defined as FPG of 110-125mg/dL. So I can agree that isolated IFG is less than isolated IGT.

In your studies, IFG was defined as FPG of 100-125mg/dL. I can guess isolated IFG is more common than isolated IGT.

Our response:

Dear Dr. Kim, thank you very much for your positive and meticulous comments. Our response to your comment is listed as following.

We have carefully considered your comment. Consistently, in two recently published meta-analyses, it is found that the global average proportion of isolated IFG was higher than that of isolated IGT, when using the ADA IFG classification with FPG cut-off values of 5.6-6.9 mmol/L (Yip WCY, et al. *Nutrients* 2017 Nov 22;9(11). pii: E1273. Huang Y, et al. *BMJ* 2016 Nov 23;355:i5953). A national cross-sectional study conducted in China found that according to the ADA criteria, the incidence of isolated IFG was greater than that of isolated IGT across all the age groups (Xu Y, et al. *JAMA* 2013 Sep 4;310(9):948-59). However, after reviewing more references and combining our previous work, we think whether the prevalence of isolated IFG more common than that of isolated IGT is still controversial, at least in the Chinese population. A significant body of evidences shows that no matter in a national or regional study, based on the ADA criteria, higher prevalence for isolated IGT rather than isolated IFG was detected among Chinese adults (Xiao J, et al. *J Gerontol A Biol Sci Med Sci* 2014 Apr;69(4):463-70. Bao C, et al. *PLoS One* 2015 Mar 18;10(3):e0119510. Liu W, et al. *Arq Bras Endocrinol Metabol* 2014 Oct;58(7):715-23. Li Y, et al. *Medical Journal of Chinese People's Liberation Army* 2010;1:74-8 (in Chinese)). In conclusion, these inconsistent findings did not differ in genders, ages, or ethnicities. The national surveys mentioned above only showed the overall pre-diabetes incidences nationwide, which did not display the specific glucose metabolic characteristics of each center/area respectively. It is suggested that the pre-diabetes prevalence may be distinct in different areas/regions in China.

In consistent with our current findings, our previous work conducted in southwest China, using the same ADA definition of IFG, demonstrated that the incidence of isolated IFG was less common than that of isolated IGT (Lü Q, et al. *Diabetes Res Clin Pract* 2009 Jun;84(3):319-24. Huang L, et al. *Peptides* 2014 Oct;60:86-94). Moreover, a same result was observed in a survey conducted in Chongqing (Zhang S, et al. *Chin J Diabetes* 2006;14(1):43-6 (in Chinese)). These studies implied that maybe in southwest China, isolated IGT more common than isolated IFG.

IFG occurs due to hepatic insulin resistance and IGT results from insulin resistance in skeletal muscle and tissues. Lack of glucose clearance capacity after meals could lead to IGT. Polished rice and refined wheat, with high glycemic index and glucose load values, aggravates the postprandial glucose burden. Lack of exercise causes skeletal muscle insulin resistance, which further contributes to IGT. The above dietary habit and sedentary lifestyle are widespread in southwest China.

Lifestyle, economic level, and public health system, which are closely associated with pre-diabetes incidences, are quite disparate in different areas/regions. Therefore, it is reasonable to believe that the prevalence of isolated IFG or/and isolated IGT could be distinct from different areas/regions. Further studies are needed to explore and compare the metabolic characteristics of Chinese adults in different areas.

At last, dear Dr. Kim, we sincerely appreciated your time and efforts for your constructive and meticulous advice.